# Down-Regulation of AKT Proteins Slows the Growth of Mutant-KRAS Pancreatic Tumors

**DOI:** 10.3390/cells13121061

**Published:** 2024-06-19

**Authors:** Chuankai Chen, Ya-Ping Jiang, Inchul You, Nathanael S. Gray, Richard Z. Lin

**Affiliations:** 1Department of Physiology & Biophysics, Stony Brook University, Stony Brook, NY 11794, USA; chuankai.chen@stonybrook.edu (C.C.); yaping.usa@gmail.com (Y.-P.J.); 2Graduate Program in Genetics, Stony Brook University, Stony Brook, NY 11790, USA; 3Department of Chemical and Systems Biology, ChEM-H, and Stanford Cancer Institute, Stanford School of Medicine, Stanford University, Stanford, CA 94305, USA; inchulyou01@gmail.com (I.Y.); nsgray01@stanford.edu (N.S.G.); 4Northport VA Medical Center, Northport, NY 11768, USA

**Keywords:** pancreatic cancer, AKT, IGF-1, cholesterol metabolism

## Abstract

Serine/threonine kinase AKT isoforms play a well-established role in cell metabolism and growth. Most pancreatic adenocarcinomas (PDACs) harbor activation mutations of KRAS, which activates the PI3K/AKT signaling pathway. However, AKT inhibitors are not effective in the treatment of pancreatic cancer. To better understand the role of AKT signaling in mutant-KRAS pancreatic tumors, this study utilized proteolysis-targeting chimeras (PROTACs) and CRISPR-Cas9-genome editing to investigate AKT proteins. The PROTAC down-regulation of AKT proteins markedly slowed the growth of three pancreatic tumor cell lines harboring mutant KRAS. In contrast, the inhibition of AKT kinase activity alone had very little effect on the growth of these cell lines. The concurrent genetic deletion of all AKT isoforms (AKT1, AKT2, and AKT3) in the KPC (*Kras*^G12D^; *Trp53*^R172H^; *Pdx1-Cre*) pancreatic cancer cell line also dramatically slowed its growth in vitro and when orthotopically implanted in syngeneic mice. Surprisingly, insulin-like growth factor-1 (IGF-1), but not epidermal growth factor (EGF), restored KPC cell growth in serum-deprived conditions, and the IGF-1 growth stimulation effect was AKT-dependent. The RNA-seq analysis of AKT1/2/3-deficient KPC cells suggested that reduced cholesterol synthesis may be responsible for the decreased response to IGF-1 stimulation. These results indicate that the presence of all three AKT isoforms supports pancreatic tumor cell growth, and the pharmacological degradation of AKT proteins may be more effective than AKT catalytic inhibitors for treating pancreatic cancer.

## 1. Introduction

Pancreatic ductal adenocarcinoma (PDAC) is a highly fatal disease with a 5-year survival rate of ~10% [1]. Mutant KRAS is found in more than 90% of cases and has been established as a driving mutation in PDAC. KRAS and its best-known MAPK effector signaling pathway have been studied extensively for PDAC biology and treatment [2,3]. PI3K is another effector signaling pathway downstream of KRAS. We and others have shown that this signaling pathway is important for the initiation and progression of PDAC [4,5,6,7]. Downstream of PI3K is the serine/threonine kinase AKT, which has three paralogous genes, i.e., *AKT1*, *AKT2*, and *AKT3* with distinct expression patterns in normal tissue. AKT1 is ubiquitously expressed; AKT2 is enriched in insulin-responsive tissues, i.e., liver, fat tissue, and muscle; and AKT3 expression is enriched in the brain [8]. AKTs are generally considered the major effectors of PI3Ks in virtually all physiological and pathological tissues, including various cancers [8]. However, currently available AKT kinase inhibitors were not found to be effective in clinical trials for PDAC patients [9].

PROATCs (henceforth referred to as degraders) represent a new strategy of targeting kinases by engaging proteasomes to degrade kinases [10]. AKT degraders, especially MS21 and INY-05-040, have been developed recently. MS21 has proven to be effective in reducing cell growth in mainly breast cancers with PI3K/PTEN mutations but not KRAS mutations [11]. INY-05-040 has been shown to be effective for breast cancer lines with a low-baseline activation of stress MAPK pathway [12]. In this study, we took advantage of the latest AKT protein degrader INY-05-040 to study pancreatic tumor growth. In addition, we utilized CRISPR-Cas9 to genetically delete all three AKT isoforms individually and in combination to investigate the effects on PDAC progression in vitro and in vivo. Our results demonstrated that AKT proteins play a critical role in promoting pancreatic cancer progression whereas inhibition of AKT kinase activity alone has little effect. These results suggest that AKT protein degradation could be a therapeutic strategy for PDAC patients.

## 2. Materials and Methods

### 2.1. Cell Lines

FC1245 was obtained from David Tuveson (Cold Spring Harbor Laboratory, Cold Spring Harbor, NY, USA), and PANC-1 was purchased from ATCC. Both lines were cultured in DMEM supplemented with 10% fetal bovine serum (FBS) and 1% Penstrep. The patient-derived pancreatic cancer cell line UM5 was a gift from Costas Lyssiotis (University of Michigan, Ann Arbor, MI, USA) and cultured in RPMI with 10% FBS and 1% Penstrep.

### 2.2. AKT Degrader and Inhibitor Treatment

The AKT degrader INY-05-040 and its control compound INY-05-041 were provided by Nathaniel Gray (Stanford University, Stanford, CA, USA). GDC0068 was purchased from Selleckchem, Houston, TX, USA. To find the optimal working concentration of the degrader, a titration experiment was performed in FC1245 cells (Appendix A). In brief, KPC cells were treated with increasing concentrations of the AKT degrader INY-05-040 or the control compound INY-05-041. Because the AKT degradation efficacy plateaued after 500 nM, it was thus chosen for the colony formation assay of FC1245 cells. In PANC-1 cells, 500 nM degraders could also degrade AKT effectively. In UM5 cells, since 500 nM INY-05-040 had only minimal effects, 1 μM was used for the colony formation assay instead.

### 2.3. Crispr-Cas9-Mediated Genome Editing

The CRISPR-Cas9 constructs targeting murine *Akt1*, *Akt2*, and *Akt3* were purchased from Santa Cruz Technology, Dallas, TX, USA. *Akt1*KO, *Akt2*KO, and *Akt3*KO cell lines were generated by transfecting parental KPC cells with constructs targeting the respective *Akt* genes. *Akt1/3*KO and *Akt 2/3*KO cell lines were generated by transfecting *Akt3*KO cells with *Akt1*-targeting or *Akt2*-targeting CRISPR constructs followed by cloning. *Akt1/2/3*KO cell lines were generated by transfecting *Akt2/3* with *Akt1* targeting CRIPSR plasmid and cloning.

### 2.4. Immunoblotting

The immunoblotting method has been described before [6]. In brief, cells were harvested in RIPA buffer supplemented with 16 mM sodium fluoride, 32 mM β-glycerophosphate, 6.4 mM sodium pyrophosphate, 0.8 mM PMSF, 4 mM orthovanadate, and 1:10 protease inhibitor cocktail p8340 (Sigma, Marlborough, MA, USA). Immunoblots were performed with Bio-Rad Mini Protean Tetra Cell Gel Electrophoresis and a Trans-Blot^®^ SD Semi-Dry Transfer Cell system. Blots were incubated with primary antibodies in 5% non-fat milk or 5% BSA overnight and secondary HRP-conjugated antibodies. Blots were imaged and visualized by a Protein Simple FluorChem system. The antibodies used were: total AKT (Cell Signaling, Dambers, MA, USA, #2920), p-AKT s473 (Cell Signaling, #4060), HSP90 (Cell Signaling, #4874), p-S6 (Cell Signaling, #2215), p-PRAS40 (Cell Signaling, #2997), AKT1 (Cell Signaling, #2967), AKT2 (Cell Signaling, #3063), AKT3 (Cell Signaling, #3788), ACTIN (Cell Signaling, #3700), PRAS40 (Cell Signaling, #2961), p-GSK3beta (Cell Signaling, #9336), GSK3beta (Cell Signaling, #9315), p-MEK1/2 (Cell Signaling, #9121), MEK1 (Cell Signaling, #2352), p-ERK1/2 (Cell Signaling, #4376), and ERK2 (Santa Cruz Technology, Dallas, TX, USA, sc-154).

### 2.5. Cell Counting and Colony Formation Assay

For the cell counting experiments, 50,000 cells were seeded into each well of six-well plates and counted over the 4-day incubation period. The experiments were performed three times with cells of different passages.

In the colony formation assay when cells were treated with either the AKT degrader or inhibitor, 50 cells were seeded into each well of the 6-well plates, and media were replaced every 4 days until distinct colonies were formed. The cells were then stained with crystal violet and formalin. A 5% FBS medium was used for colony formation assays involving drug treatment. For colony formation assays without drug treatment, media containing 10% FBS were used. Images were captured with a ProteinSimple Flurochem instrument, and colony areas were quantified by ImageJ (1.53t). A grayscale range of 156–255 was used as the image threshold and the colonies with areas larger than one pixel were counted.

For the colony formation assay with growth factor supplementation, 10,000 cells were seeded to each well of a 96-well plate; on day 2, the original media were replaced with DMEM with PBS, DMEM containing 100 ng/mL EGF or 100 ng/mL IGF-1 or 100 ng/mL EGF plus 100 ng/mL IGF-1, or 10% FBS, and the media were replaced every day to minimize the effects of autocrine growth factors. Cells were stained on day 5 and quantified as described above, and the grayscale range 182-255 was used as the image threshold.

### 2.6. Three-Dimensional Culture

As described previously [6], 1000 cells were suspended in 10% FBS DMEM with 0.24% methylcellulose and cultured by the hanging drop method. Specifically, cells were transferred to the inner surface of the lip of a 24-well plate and PBS was added to the wells to prevent evaporation. Cells were cultured for 6 days before pictures were taken with a brightfield microscope at 40× magnification.

### 2.7. Apoptosis Assay and Cell Cycle Analysis

Apoptosis assays were performed according to the manufacturer’s instructions (Invitrogen, Waltham, MA, USA, V13242). In brief, 50,000 cells were seeded in each well of 6-well plates, cultured in DMEM with 10% FBS for 4 days, harvested, washed, and resuspended to be about 1 million cells per microliter, and incubated with Annexin V FITC and propidium iodide (PI) for about 15 min before immediate flow cytometric analysis. For cell cycle analysis, DNA staining was performed according to the manufacturer’s instructions (Invitrogen, P3566). In brief, 50,000 cells were seeded in each well of 6-well plates in duplicates, cultured in DMEM with 10% FBS for 2 days, then harvested and fixed in −20 alcohol for 15 min, counterstained with 3 μM PI solution, and processed on a flow cytometer. The results were analyzed by the FlowJo software package (v10.8.1).

### 2.8. Orthotopic Implantation and IVIS Imaging

Cancer cells were transduced with retroviruses containing luciferase-encoding genes and selected by G418 or blasticidin S. C57BL/6J mice (#000664) were purchased from Jackson Laboratories. The orthotopic implantation surgeries have been described previously [6]. In brief, cells were trypsinized and washed twice with PBS, then counted with a cell counter to prepare 5 × 10^5^ cells in 30 μL of PBS per mouse for injection. Mice were anesthetized with a combination of 100 mg/kg ketamine and 10 mg/kg xylazine, followed by a small vertical incision made over the left lateral abdominal area. The pancreas was then located with the aid of a light microscope, and the injection was made at the head of the pancreas by a sterile Hamilton syringe with a 27-gauge needle. After sutures, the mice were given an intraperitoneal injection of 2 mg/kg ketorolac. The tumor growth was monitored by the IVIS in vivo imaging system as previously described [6].

### 2.9. RNA-Seq Analysis

Parental KPC and Akt1/2/3KO of three different passages were seeded to 6-well plates and cultured in DMEM with 10% FBS for 3 days. The RNAs of triplicate samples were harvested with an RNeasy kit (Qiagen, Hilden, Germany) and sent to Novogene for library construction and paired-end sequencing. Data were processed in R by the package Rsubread (v2.12.3) for alignment and feature counting, and by the package edgeR (v3.40.2) for differential expression analysis. Then, pathway enrichment analysis was conducted with the desktop version of GSEA software (v4.2.3). Pathway clustering and visualization were performed using EnrichmentMap in the Cytoscape software package (v3.10.1).

### 2.10. Amplex Red Cholesterol Measurement Assay

A total of 20,000 cells were seeded in each well of 96-well plates in DMEM with 2.5% FBS on day 1 and the media were replaced with 10% FBS or serum-free DMEM on day 2, and the Amplex assay was performed on day 4 according to the manufacturer’s instructions.

### 2.11. RPPA

In brief, 200,000 cells were seeded in 6 cm plates in duplicates and cultured for 3 days in 10% FBS DMEM. Cell lysates processed with lysis buffer provided by the MD Anderson RPPA Core were sent for analysis by the facility. The linearized L2 data were used in our analysis.

## 3. Results

### 3.1. AKT Degrader Was Superior to AKT Kinase Inhibitor in Slowing Pancreatic Cancer Cell Growth

In recent publications, You, et al. developed new compounds that could recruit ubiquitin ligase E3 to down-regulate AKT proteins through proteasome degradation [12,13]. We first compared the AKT degrader (INY-05-040) versus the clinically tested AKT catalytic inhibitor (GDC0068 or ipatasertib) in the human pancreatic cancer cell line PANC-1, the low-passage patient-derived pancreatic cancer cell UM5, and a pancreatic cancer cell line derived from the KPC mouse model (*Kras*^G12D^; *Trp53*^R172H^; *Pdx1*-Cre). Although the AKT inhibitor had a minimal effect on the growth of all three cell lines as shown by the colony formation assay, the AKT degrader dramatically reduced colony formation in all three cell lines (Figure 1A).

The AKT degrader effectively reduced pan-AKT protein levels in PANC-1 and KPC cells, but surprisingly, only minimally reduced pan-AKT protein levels in UM5 cells. Expectedly, the AKT inhibitor caused a small but consistent reduction in the phosphorylation levels of downstream target proteins, i.e., S6 in PANC-1 and UM5, and PRAS40 in KPC (Figure 1B–D). Interestingly, the AKT inhibitor, but not the degrader treatment, induced compensatory AKT hyperphosphorylation in all three cell lines (Figure 1B–D).

### 3.2. Generation of KPC Cell Lines Deficient in AKT Isoforms by CRISPR-Cas9 Genome Editing

The role of AKT isoforms in pancreatic cancer remains ill-defined [14,15,16,17,18,19]. Both INY-05-040 and GDC0068 target all three AKT isoforms, and there is always a potential for off-target effects from pharmacological agents [20,21]. To investigate whether individual AKT isoforms alone or in combination play a role in promoting PDAC cell growth, we genetically deleted the AKT isoforms individually and in all possible combinations using CRISPR-Cas9-mediated genome editing. In all, we generated seven knockout cell lines from the parental KPC cells. We first generated single-gene knockouts, i.e., *Akt1*KO, *Akt2*KO, and *Akt3*KO, from which we then generated double-gene knockouts, i.e., *Akt1/3*KO, *Akt2/3*KO, and *Akt1/2*KO, and then finally the triple-gene knockout *Akt1/2/3*KO (Figure 2A).

The knockout efficiency was confirmed by immunoblotting (Figure 2B). AKT1 is the most abundant AKT protein in KPC cells, followed by AKT3, and then AKT2 (Figure 2B). AKT3 has been reported to be expressed in various cancers, including breast cancers, ovarian cancers, and gliomas [22,23,24,25,26,27,28]. *AKT3* mRNA expression has been reported previously in pancreatic cancer cells [16,17,29,30,31]. To our knowledge, this was the first report of AKT3 protein expression in a pancreatic cancer cell line confirmed by gene editing. Surprisingly, AKT3 contributed the most to the phosphorylation of the downstream targets (PRAS40 and GSK3β) in KPC cells (Figure 2C), whereas the genetic deletion of AKT1 led to smaller decreases in the phosphorylations of PRAS40 and GSK3β (Figure 2C). The genetic deletion of AKT2 had little effect on the phosphorylation of those two downstream targets (Figure 2C). It is important to note that the three AKT isoforms play non-redundant roles in signaling downstream since the deletion of all isoforms was necessary to decrease the phosphorylation of those two target proteins to the lowest level (Figure 2C, see *Akt1/2/3*KO results). The ERK/MPAK pathway is another important effector pathway of mutant KRAS in PDAC, and it can cross-talk with the PI3K/AKT signaling pathway [8]. However, ablation of all three AKT isoforms (see *Akt1/2/3*KO results) did not decrease but instead increased MAPK signaling (Figure 2D). Therefore, phenotypic changes observed in *Akt1/2/3*KO cancer cells were unlikely due to the down-regulation of MAPK signaling.

### 3.3. Genetic Deletion of Akt1/2/3 Significantly Impeded In Vitro PDAC Cell Growth

Despite this compensatory upregulation of ERK/MAPK signaling in *Akt1/2/3*KO cells, this triple-gene-deleted cell line had the slowest growth rate as compared to all the other knockout cell lines as measured by cell counting (Figure 3A). Furthermore, we also found a discrepancy between the contribution of the kinase activity of AKT isoforms to downstream signaling and their impact on cell growth. For example, the loss of AKT1 plus AKT2 reduced cell growth more significantly than other double- or single-AKT-gene-deleted cell lines, but PRAS40 and GSK3β phosphorylations were minimally reduced in *Akt1/2*KO cells (compare Figure 2C to Figure 3A). In contrast, the *Akt2/3*KO cells had a very significantly decreased phosphorylation of downstream effectors (Figure 2C), at a level comparable to those of *Akt1/2/3*KO cells, but surprisingly minimal cell growth reduction (Figure 3A). We confirmed that all knockout cell lines had similar apoptotic rates (Appendix A). Taken together, these results suggest that (1) individual AKT isoforms are redundant in promoting KPC in vitro growth; (2) AKT1 was sufficient for sustaining fast KPC growth, consistent with the growth-promoting role of this isoform in most non-cancer cell types [8]; (3) whereas AKT3 (and supported by AKT2) contributed more to the phosphorylation of the downstream effectors, PRAS40 and GSK3β, they were largely dispensable for KPC cell growth in the presence of AKT1.

To further support our observation of the in vitro cell growth phenotype, we next performed a colony formation assay on these gene-deleted cell lines. Cells were grown for 7 days prior to evaluation. There was a distinct decrease in the colony numbers of *Akt1/2/3*KO cells compared to parental KPC cells (Figure 3B). More importantly, the colony sizes were dramatically different in the AKT gene-deleted cell lines, especially *Akt1/2/3*KO colonies, which were the smallest (Figure 3B). These results were consistent with the AKT degrader treatment experiments (see Figure 1A). As expected, AKT degrader treatment did not further slow the growth of *Akt1/2/3*KO cells, indicating that AKT degrader INY-05-040 was highly selective (Appendix A).

AKT directly phosphorylates and regulates cycle regulators to control cell cycle progression [32]. However, we did not find that *Akt1/2/3*KO cells had a higher percentage of cells in the G1 or G2 phase; there was a slightly higher percentage of *Akt1/2/3*KO cells in the S phase (Appendix A). Taken together, these results suggest that *Akt1/2/3*KO cells did not have a significant degree of cell cycle arrest.

### 3.4. Genetic Ablation of Akt1/2/3 Slowed KPC Pancreatic Tumor Progression in a Syngeneic Orthotopic Implantation Mouse Model

To study how the loss of AKT isoforms affects tumor growth in vivo, we orthotopically implanted the parental KPC and seven AKT gene-deleted cell lines in the pancreas of syngenetic C57BL/6J mice. Tumor progression was monitored by IVIS imaging and animal survival curves were generated. Consistent with in vitro growth results, the *Akt1/2/3*KO pancreatic tumors progressed the slowest compared to all the other cell lines (Figure 4A,B), and the host mice implanted with *Akt1/2/3*KO cells had the longest median survival of 86 days. This was a significant improvement from the mice implanted with the parental KPC cell line, which had a median survival of 15.5 days (Figure 4C). Furthermore, while the mice with all other KO cell lines died eventually due to tumor progression, ~36% of the host mice implanted with *Akt1/2/3*KO cells experienced spontaneous tumor regression, leading to long-term tumor-free survival (Appendix A). We confirmed that *Akt1/2/3*KO cells were capable of anchorage-independent growth by a 3D culture; therefore, it was not the cause of slower tumor growth in vivo (Appendix A). Interestingly, *Akt1/2*KO pancreatic tumors also progressed relatively slowly leading to prolonged host animal survival with a median of 47 days; however, there were no long-term tumor-free survivors (Figure 4A,B). The implantation of all other single- or double-AKT-gene-deleted cell lines led to rapid tumor growth and host animal death with median survival times similar to that of the implantation of the parental KPC cell line (Figure 4A,B).

### 3.5. Growth Effects of IGF-1-AKT Signaling in the KPC Cell Line and Role of Cholesterol Metabolism

Growth factor signaling through receptor tyrosine kinases (RTKs) has been shown to be critical for PDAC tumorigenesis and growth in model systems [33,34,35,36]. Their role in tumor progression is also highly suspected in pancreatic cancer patients [37]. Since AKT is a common downstream effector of many growth factor signaling pathways in numerous cell types, we thus asked whether AKT also mediates growth factor signaling in the KPC cell line. We stimulated parental KPC and *Akt1/2/3*KO cells with either IGF-1 or EGF alone or in combination in colony formation assays. We found that IGF-1 (with or without EGF) but not EGF alone could promote KPC cell growth in the absence of serum (Figure 5A). Importantly, the IGF-1 growth stimulatory effect was completely eliminated in the *AKT1/2/3*KO cell line (Figure 5A). The lack of effect of EGF stimulation on KPC cells under this serum-free condition is consistent with our previous observation that the *Egfr*-null KPC line has comparable in vitro and in vivo growth rates to the parental KPC cell line [6]. Our experimental finding also directly supported the model proposed by Tape, C.J., et al. based on a proteomics analysis, in which the IGF1R-AKT axis activation of pancreatic cancer cells is an important regulator of PDAC growth [38].

To further characterize the *Akt1/2/3*KO cell line and find additional growth regulators in the AKT pathway, we performed RNA-seq analysis on parental KPC and *Akt1/2/3*KO cell lines. There were dramatic differences between the transcriptomes of KPC and *Akt1/2/3*KO cells (Figure 5B). We performed a GSEA pathway enrichment analysis followed by clustering and visualization in Cytoscape (EnrichmentMap). The results suggested that the cholesterol metabolism was significantly altered in *Akt1/2/3*KO cells (Figure 5C). We found that key genes in the mevalonate pathway for cholesterol synthesis were significantly down-regulated. The mRNA expressions of *Hmgcr* and *Hmgcs1*, encoding two key enzymes catalyzing the rate-limiting steps of the mevalonate pathway, were decreased in *Akt1/2/3*KO cells (Figure 5D). *Ldlr,* the encoding cell surface receptor responsible for cholesterol-rich low-density lipoprotein uptake, was also significantly decreased (Figure 5D). Interestingly, *Srebf2* encoding a master transcriptional regulator of cholesterol synthesis was also down-regulated (Figure 5D). Consistent with the RNA-seq findings, we found that KPC cells, but not *Akt1/2/3*KO cells, could increase cholesterol production upon serum deprivation (Figure 5E). Previous studies have shown that cell membrane cholesterol content can affect growth factor receptor signaling, including the IGF-1 receptor (IGF-1R) [39,40,41,42,43,44,45,46]. This potentially is one mechanism contributing to the reduced responsiveness of *Akt1/2/3*KO cells to IGF-1 growth stimulation [39,40,41,42,43,44,45,46].

## 4. Discussion

AKT inhibitors as single-agent treatments have not been effective in pancreatic cancer clinical trials [47,48,49]. One possible reason for these results is incomplete AKT kinase activity inhibition using currently available compounds. Recently, there has been the development of a new strategy using PROTACs for targeting kinases including AKT by co-opting proteasome-mediated degradation [10]. There have been many clinical trials showing PROTACs targeting various proteins for cancer treatment [50]. The PROTAC ARV-471 targeting estrogen receptor (ER) has entered phase III clinical trial for the treatment of ER+/HER2− locally advanced or metastatic breast cancer. Clinical trials using AKT degraders including INY-05-040 have not been reported, but some of the AKT degraders, especially INY-05-040 and MS21, have been tested in xenograft mouse models [12,51]. MS21 has been demonstrated to have more efficacies than its parental drug in reducing tumor volumes in prostate and breast cancer xenograft mouse models [51].

AKT degraders were designed based on AKT inhibitors, but they possess additional chemical groups so that they can engage ubiquitin ligases leading to the degradation of target proteins through proteasomes [11,12,13,51,52,53,54]. A number of studies have found that this novel class of compounds is more effective than their parental AKT inhibitors in suppressing downstream signaling and reducing cancer cell growth. They were particularly effective against breast and prostate cancer cell lines with HER2/PI3K/AKT mutations [11,12]. The investigators believed that these compounds could induce a more extensive and sustained AKT signaling suppression. Indeed, we found that AKT degrader treatment reduced mouse and human pancreatic cancer cell lines with apparently minimal off-target effects. However, we did not observe a greater cell signaling suppression by the AKT degrader INY-05-040 than the AKT inhibitor GDC0068. In light of the report of the kinase-independent function of AKT in breast cancer cell lines [55], it remains possible that some or all three AKT isoforms have additional functions in pancreatic cancer cells other than their catalytic activity.

AKT isoforms play distinct and sometimes opposing roles in breast cancer initiation versus differentiation [56,57,58,59]. In pancreatic cancer, we discovered that AKT isoforms are major regulators of pancreatic cancer growth. Specifically, AKT1 appears to be the major regulator of KPC cell growth whereas AKT2 and AKT3 are more important for the phosphorylation of the downstream targets GSK3β and PRAS40. Expectedly, *Akt1/2/3*KO cells have no AKT protein and dramatically reduced AKT downstream signaling (assessed by GSK3β and PRAS40 phosphorylation). However, for the single and double AKT knockout cell lines, we did not find a correlation between AKT downstream signaling or total AKT protein levels to their growth rates. These results suggest that (1) other critical kinase targets of AKT, such as mTORC2 and/or FOXO, play a role in regulating KPC growth; (2) the subcellular localization of AKT isoforms may be important for their activity and function; or (3) AKT proteins may contribute to KPC cell growth in a kinase-independent manner [55]. Indeed, all three possible mechanisms may be contributing to their effects on cell growth.

Tumor–stromal interactions have been intensively studied in PDAC [60,61]. IGF-1, for example, can be produced by activated stromal fibroblasts and may contribute to cancer cell growth [37]. Diabetes mellitus is a risk factor for developing pancreatic cancer [62]. High levels of insulin found in diabetic patients can also activate the IGF1 receptor and this may contribute to cancer cell growth [37]. Our data also demonstrated that IGF-1 is an important regulator of KPC cell growth in vitro and importantly identified that AKTs critically mediate this effect. The efficacy of targeting the IGF-1 receptor with the small molecule linsitinib or antagonist antibody teprotumumab for pancreatic cancer treatment may be tested in PDAC mouse models, such as the syngeneic orthotopic implantation model used in our study. Additional in vitro and in vivo studies using *Igf1r*-null PDAC cancer cell lines may also be informative.

KPC cells in serum-replete complete media still grew faster than in IGF-1-supplemented serum-free media (see Figure 5A), suggesting there are additional factors in serum that can promote KPC cell growth. Tape, C.J., et al. reported that stromal-sourced factors can activate AKT through the cell-surface receptor AXL [38]. Future experiments should investigate if targeting AXL and other cell surface receptors can inhibit pancreatic cancer progression. These targets may be more clinically “druggable” than AKT.

AKT has been shown to regulate lipid metabolism including cholesterol synthesis in immortalized human retinal pigment epithelial cells, HEK 293, hepatocytes, and various cancer cells [45,63,64,65,66], and this metabolic regulation is conserved from mammals to flies [64]. Our RNA-seq analysis indicated that *Akt1/2/3*KO cells down-regulate key genes responsible for cholesterol synthesis and uptake. We found that AKTs regulate the expression of *Srebf2* but not *Srebf1* (Figure 5D and Appendix A). Currently, it is unknown if the two major lipid metabolism transcription factors, i.e., SREBF1 and SREBF2, are differentially regulated by AKT [45]. Lastly, we showed that although KPC and *Akt1/2/3*KO cells had similar cholesterol levels when cultured in serum-replete media, *Akt1/2/3*KO cells could not increase cholesterol production when placed in serum-free media, a stressful condition similar to the hypovascularized and low-nutrition in vivo environment [67,68]. Interestingly, Erickson, et al. have recently shown that AKT degradation but not kinase inhibition modulates expressions of genes critical for cholesterol metabolism in breast cancer cells, which is highly supportive of our results in pancreatic cancer cells [12].

Cholesterol is an important membrane component of cancer cells, and cholesterol derivatives promote cancer progression by modulating cell signaling [69] and contributing to an immunosuppressive environment [70]. The de novo inhibition of cholesterol synthesis, uptake, or storage has been shown to greatly impede pancreatic cancer growth [71,72,73,74]. Additionally, there have been many reports supporting the concept that cholesterol-rich lipid rafts modulate the sensitivity of insulin and IGF-1 receptors to their respective agonists in various cell types [39,40,41,42,43,44,45]. Delle Bovi demonstrated that IGF-1 receptor activation is dependent on cytoplasmic cholesterol content and IGF-1 receptor sensitivity to IGF-1, which is augmented by the membrane reconstitution of cholesterol [46]. Our results suggest that the loss of AKTs in pancreatic cancer cells may down-regulate membrane cholesterol levels to decrease the responsiveness to IGF-1. *Akt1/2/3*KO cells showed upregulated signature epithelial-to-mesenchymal transition (EMT) gene expression and down-regulated classical PDAC gene expression (see Appendix A), which are associated with PDAC cell differentiation [75]. These results support the report by Gabitova-Cornell, et al. that showed disrupting cholesterol synthesis initiates the EMT program in pancreatic cancer cells [71].

The in vivo progression of the *Akt*-deficient KPC cell lines generally correlated with in vitro growth. There was a significant improvement in the survival of *Akt1/2/3*KO-cell-implanted mice compared to those of parental-KPC-cell-implanted mice. The tumors in ~36% of the mice regressed completely and the mice remained tumor-free until the end of the experiment. This was partially consistent with our report of *Pik3ca*KO KPC, where we observed that *Pi3kca* knockout led to complete tumor regression and 100% mice survival [6]. The difference in the penetrance of the phenotype suggests that other PI3KCA targets in addition to AKT1/2/3 play a mediating role in PI3K’s regulation of pancreatic cancer immunity.

## 5. Conclusions

The AKT degrader INY-05-040 outperformed the catalytic inhibitor GDC0068 in suppressing pancreatic cancer cell growth, even though both compounds had comparable suppressive effects on downstream signaling. The genetic deletion of AKT isoforms in the KPC cell line indicated that AKT isoforms have redundant roles in regulating pancreatic cancer cell growth and ablation of all three AKT proteins is needed to reduce cancer growth to the lowest level. However, our results also indicated these isoforms have distinct roles. AKT1 appears to contribute more to pancreatic cancer growth whereas AKT2 and AKT3 contribute more to phosphorylating downstream targets. Lastly, cholesterol metabolism and IGF-1-stimulated growth in pancreatic cancer cells were both regulated by AKT proteins. Future studies that investigate the efficacy of AKT degraders in pancreatic cancer animal models and then clinical trials could be beneficial for developing better treatments for this deadly cancer.

## Figures and Tables

**Figure 1 cells-13-01061-f001:**
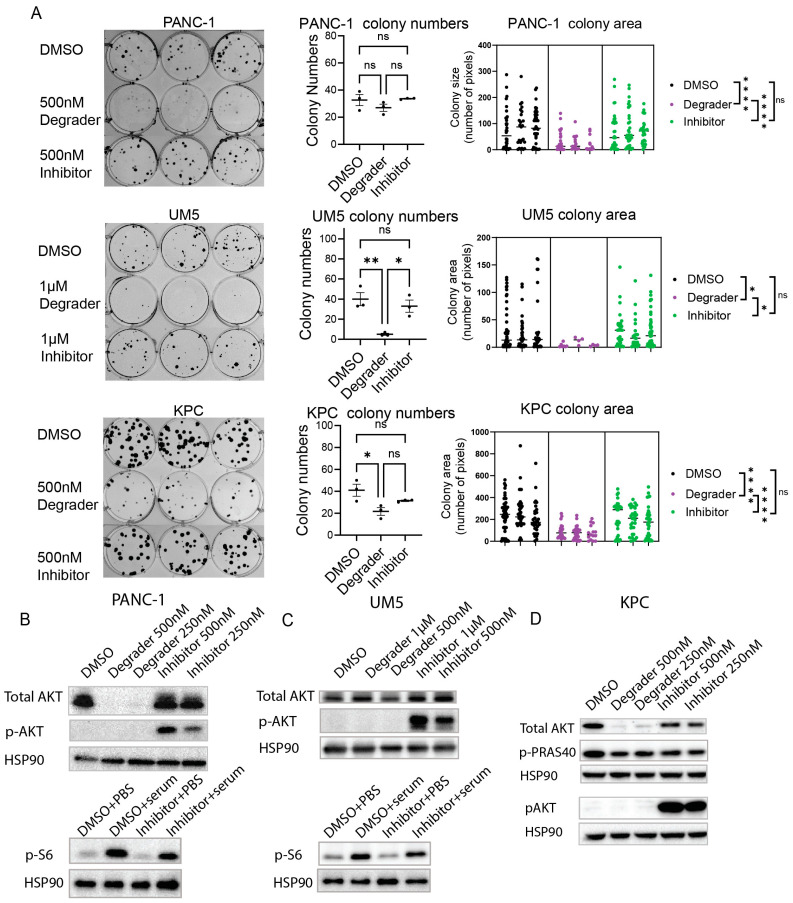
The AKT degrader outcompeted AKT inhibitors in impeding cell growth. (**A**) The colony formation assay compared the effect on colony formation of the AKT degrader INY-05-040 and inhibitor GDC0068 in the human pancreatic cancer cell line PANC-1, low-passage patient-derived PDAC cell UM5, and mouse PDAC cell line KPC. A total of 50 cells were seeded into each well of 6-well plates and grew in indicated conditions for 9 days, 16 days, or 20 days, respectively, before crystal violet staining. Media were replaced every 4 days once drug treatment started. Left, pictures of the colonies. Middle and right, the quantifications of colony sizes and numbers of the colony formation assay. Data were analyzed with a one-way ANOVA, followed by multiple comparisons with the Tukey’s method. ns not significant, * *p* < 0.05, ** *p* < 0.01, **** *p* < 0.0001. (**B**–**D**) Immunoblot comparing the AKT degrader and inhibitor treatment in the human pancreatic cancer cell line PANC-1, low-passage patient-derived PDAC cell UM5, and mouse KPC cell line FC1245, respectively. Cells were cultured as in (**A**).

**Figure 2 cells-13-01061-f002:**
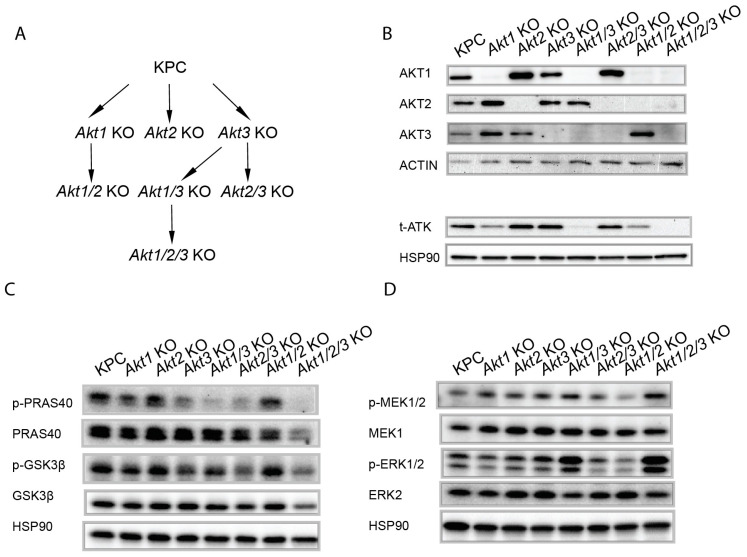
Generation of KPC cell lines deficient in *Akt* isoforms by CRISPR-Cas9 genome editing. (**A**) Schematic of the strategy to genetically delete all three *Akt* paralogous genes. (**B**) Immunoblot of AKT proteins in knockout cell lines. **(C**) Immunoblot comparing AKT signaling between KPC vs. *Akt*-deficient cell lines. (**D**) Immunoblot using indicated phospho-antibodies comparing MEK/ERK signaling between KPC vs. *Akt*-deficient cell lines. Both blots were probed with anti-HSP90 antibodies as an additional loading control.

**Figure 3 cells-13-01061-f003:**
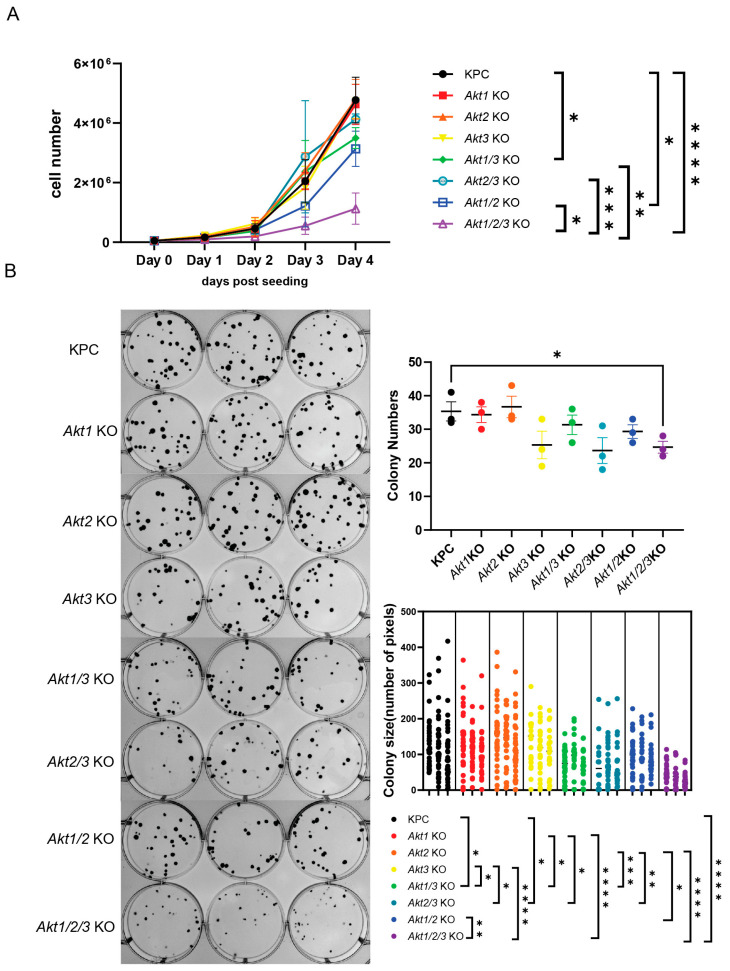
Loss of *Akt1/2/3* slowed KPC cell growth in vitro. (**A**) The growth curve of *Akt*-deficient KPC cell lines indicated *Akt1/2/3* KO cells grew the slowest. A total of 50,000 cells were seeded in each well of 6-well plates for a four-day growth assay (*n* = 3). The experiments were repeated three times with different passage numbers of the cell lines. The cell numbers were plotted as mean ± SD; the cell numbers on day 4 were analyzed by a one-way ANOVA (*p* < 0.0001). Šídák’s multiple comparisons test was conducted between every possible pair, but only comparisons with *p* < 0.05 are shown. * *p* < 0.05, ** *p* < 0.01, *** *p* < 0.001, **** *p* < 0.0001. (**B**) The colony formation assay of *Akt*-deficient cell lines. A total of 50 cells were seeded into each well of 6-well plates and grew for 7 days before evaluation. On the left are the images of the colonies of the indicated cell lines in technical replicates (*n* = 3). On the right is the quantification of the colony numbers and sizes. The colony numbers of parental KPC and *Akt1/2/3*KO cells were compared with the *t*-test. The colony sizes of all the lines were compared by a nested one-way ANOVA (*p* < 0.0001). Multiple comparisons were performed with Tukey’s method between KPC and every other cell line, but only comparisons with *p* < 0.05 are shown. * *p* < 0.05, ** *p* < 0.01, *** *p* < 0.001, **** *p* < 0.0001. See also Appendix A.

**Figure 4 cells-13-01061-f004:**
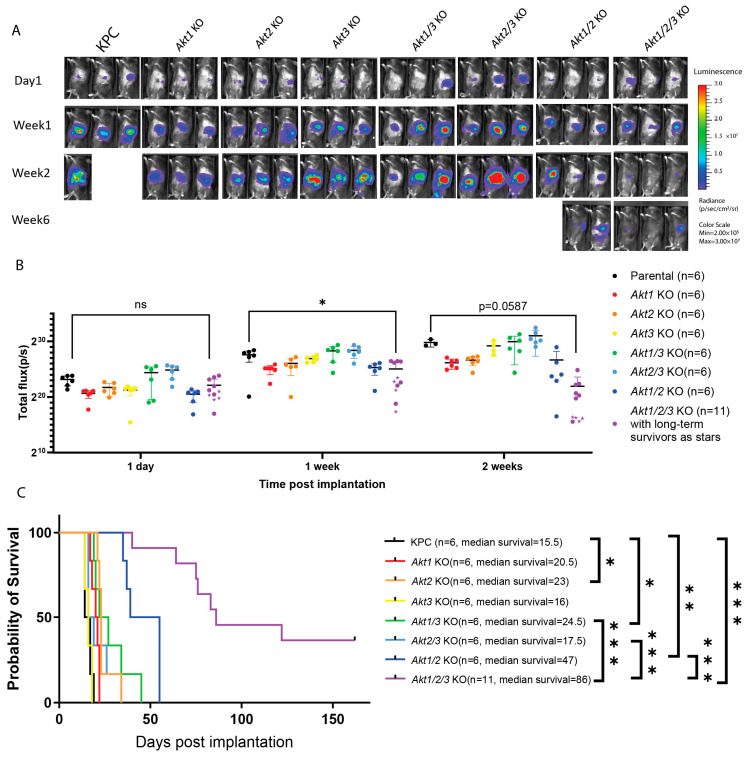
Genetic ablation of *Akt1/2/3* slowed the progression of KPC tumors in a syngeneic orthotopic implantation mouse model. (**A**) Representative IVIS images for indicated time points of surviving mice with orthotopic cancer cell implantation. B6 mice were implanted with pancreatic cancer cells of the indicated genotypes. Mice were imaged using IVIS one day post-implantation and every week thereafter. The sample sizes are labeled in (**B**). (**B**) Quantification of the total flux from IVIS data of all the mice. As there were missing values, a mixed-effects analysis was conducted between KPC and *Akt1/2/3* KO with repeated measures, followed by Tukey’s multiple comparisons test. The statistical insignificance between KPC and *Akt1/2/3* KO of the 2-week time point was in part caused by the reduced sample size due to the deaths of the mice. (**C**) The Kaplan–Meier survival curve of mice implanted with the indicated cell line. Log-rank test of the survival of animals was performed (*p* < 0.0001). The multiple comparisons between mice implanted with KPC and each of the single-, double-*Akt-*deficient cell lines, and *Akt1/2/3*KO and between *Akt1/2/3*KO and each of the double-*Akt*-deficient groups were made with Bonferroni correction, and only comparisons beyond the threshold (α_bonferroni_ = 0.05/10 = 0.005) are shown. ns not significant, * *p* < 0.005 ** *p* < 0.001, *** *p* < 0.0001.

**Figure 5 cells-13-01061-f005:**
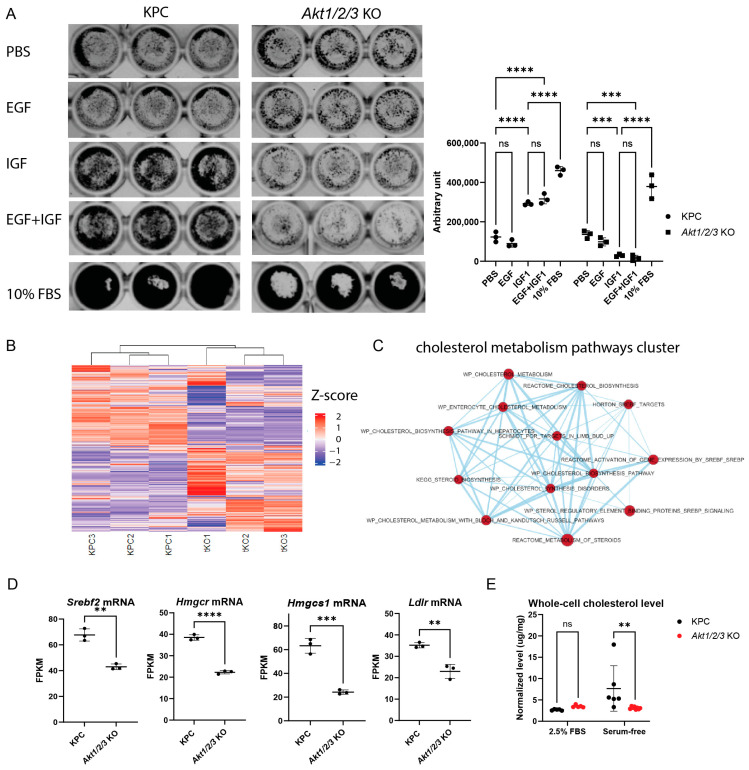
*Akt1/2/3* was required for growth-promoting IGF-1 signaling in KPC cells. (**A**) Colony formation assay of parental KPC and *Akt1/2/3*KO cells cultured in the indicated conditions. A total of 10,000 cells were seeded in a 96-well plate and treatment started on day 2 by replacing the media with either DMEM with PBS, DMEM containing 100 ng/mL EGF or 100 ng/mL IGF-1 or 100 ng/mL EGF plus 100 ng/mL IGF-1, or 10% FBS. The media were replaced every day and cells were stained with crystal violet on day 5. The quantification of the crystal-violet-positive area with Image J is shown on the right; a two-way ANOVA was performed followed by Šídák’s multiple comparisons test. *** *p* < 0.001, **** *p* < 0.001. (**B**) Heatmap of all genes by RNA-seq of KPC and *Akt1/2/3*KO (*n* = 3). The experiment was conducted with triplicates of three different passages. The color scale for the z-score is shown on the right. (**C**) GSEA pathway enrichment analysis of the differentially expressed genes between KPC and *Akt1/2/3*KO cell lines, and subsequent visualization by EnrichmentMap in Cytoscape showed a cluster of interconnected pathways related to cholesterol metabolism. (**D**) The transcript levels of key genes involved in the cholesterol synthesis pathway. *t*-tests were performed. ** *p* < 0.01, *** *p* < 0.001, **** *p* < 0.0001. *Srebp2* is a master regulator of cholesterol metabolism. *Hmgcs1* and *Hmgcr* encode the rate-limiting enzymes in the mevalonate pathway. *Ldlr* encodes the low-density lipoprotein receptor, responsible for cholesterol uptake. (**E**) Protein-mass-normalized whole-cell cholesterol levels were measured by the Amplex Red assay of KPC and *Akt1/2/3*KO cells in the indicated conditions. Two-way ANOVA with uncorrected Fisher’s LSD test. ns not significant, ** *p* < 0.01.

## Data Availability

The RNA-seq raw data have been deposited into the NCBI Sequence Read Archive (SRA) with project accession #: PRJNA1077997.

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
