# Peer review of "Down-Regulation of AKT Proteins Slows the Growth of Mutant-KRAS Pancreatic Tumors"

_cells, 2024, doi:10.3390/cells13121061_

Round 1
Reviewer 1 Report
Comments and Suggestions for Authors The author demonstrated that the AKT protein degrader INY-05-040 outperforms the catalytic AKT inhibitor in suppressing pancreatic cancer cell growth. This is a well-designed preclinical study and the data shown here include relevant findings in this field. This reviewer has one question. Is it safe to administer AKT degrader INY-05-040 in the mice model? Since down-regulation of AKT proteins in normal tissues is expected to cause severe side effects, the application of AKT protein degrader for clinical use seems to be challenging.Author Response
We thank the reviewer for the encouraging review. The AKT degrader INY-05-040 had been used in a breast cancer xenograft mouse model {Erickson, 2024 #1843} and the animals seem to tolerate the degrader up to 25 mg/kg. We have added a paragraph in the Discussion regarding previous studies with PROTAC and AKT degraders. We were not able to conduct INY-05-040 mouse experiments previously because we could not synthesize enough compound. INY-05-040 is now commercially available but would require extensive experimentation that is beyond the scope of current study.
Reviewer 2 Report
Comments and Suggestions for Authors
Chen et al., demonstrate that even though AKT1 is the most ubiquitously expressed AKT isoform, knockout and PROTAC-mediated degradation of all three isoforms are required for the progression of pancreatic cancer. Furthermore, AKT inhibitors of the shared catalytic domains are not effective as a treatment for pancreatic cancer. The authors used a sophisticated orthotopic pancreatic cancer mouse model, that further illustrates the genetic deletion of AKT1, AKT2, and AKT3. While this is an important study for those investigating better treatments for pancreatic cancer, I wonder if the authors could provide further experimental clarity on the following:
Text
1. Are there any current clinical or preclinical trials of your or any other PROTACs in vivo (mouse or human)? I think it is important to discuss even if the results are negative.
2. Though not statistically significant, AKT3 seems to do a lot. Is there anything in the literature to suggest this may be a good target for further study?
3. The description of the colony formation assays in the methods is missing in the description of Figure 5A. Also, Figure 5A is not completely clear, why use a 96-well assay with 10,000 cells in this assay and a 6-well with 50 cells for colony formation for all other assays? Is this assay soft agar? Please reconcile or redo.
Experimentally
1. Following up with comments in the text above, have you tried orthotopic experiments with PROTRAC-degraded cells, even if there is only delayed growth, that data could be important for future studies. Could also try soft agar as a more stringent colony-forming assay.
2. Have the authors examined which individual AKT was degraded in vitro in PANC1 or UM5 cells?
3. Figure 4, Panel A is hard to see.
Author Response
We thank the reviewer for the careful review and helpful comments.
Text
- We added a paragraph in the Discussion regarding PROTAC and AKT degraders in preclinical and clinical trials.
- One group reported that PANC-1 pancreatic cancer cell line and a lung cancer cell line H23, oncogenic KRAS preferentially signals to AKT3 than AKT1 or AKT2 (Geibert 2024). However, these investigators did not study the growth phenotype of these cell lines following genetic silencing of AKT isoforms.
- The experiment in Figure 5A was intended as an assay for growth factor effects. They were cultured in serum-free media (thus expecting little growth of cells) and thus we seeded more cells. All growth assays were done on standard culture plates, not in soft agar.
Experimentally
- We have not used the AKT degrader INY-05-040 in the orthotopic PDAC model. But two other AKT degraders have been tested in xenograft models; pan-AKT degrader MS21 has been demonstrated to have more efficacies than its parent drug in reducing tumor volumes in prostate and breast cancer xenograft mouse models {Yu, 2022}. Selective-AKT3 degrader 12l has been shown to be more effective than the parental drug in reducing the tumor volumes of xenograft lung cancer models {Xu, 2022}.
- We have not examined which AKT isoforms are degraded in UM5 or PANC-1 cells. In terms of isoform selectivity of the degrader INY-05-040, it has been shown to down-regulate all three isoforms in MOLT4 T lymphoblast cell line. Therefore, we suspect all three AKT isoforms in pancreatic cancer cell lines used in our study were downregulated.